# Impact of Non-Pharmaceutical Interventions on the Incidence and Treatment of Chronic Rhinosinusitis during the COVID-19 Pandemic: A Nationwide Retrospective Cohort Study

**DOI:** 10.3390/jcm12206629

**Published:** 2023-10-19

**Authors:** Chan Min Jung, Minkyung Han, Hyung-Ju Cho, Chang-Hoon Kim, Inkyung Jung, Min-Seok Rha

**Affiliations:** 1Department of Otorhinolaryngology, Gangnam Severance Hospital, Yonsei University College of Medicine, Seoul 06319, Republic of Korea; jungcm01@yuhs.ac; 2Biostatistics Collaboration Unit, Department of Biomedical Systems Informatics, Yonsei University College of Medicine, Seoul 03722, Republic of Korea; minkyunghan@yuhs.ac; 3Department of Otorhinolaryngology, Yonsei University College of Medicine, Seoul 03722, Republic of Korea; hyungjucho@yuhs.ac (H.-J.C.); entman@yuhs.ac (C.-H.K.); 4The Airway Mucus Institute, Yonsei University College of Medicine, Severance Hospital, Seoul 03722, Republic of Korea; 5The Korea Mouse Sensory Phenotyping Center, Yonsei University College of Medicine, Seoul 03722, Republic of Korea; 6Division of Biostatistics, Department of Biomedical Systems Informatics, Yonsei University College of Medicine, Seoul 03722, Republic of Korea; 7Severance Biomedical Science Institute, Yonsei University College of Medicine, Seoul 03722, Republic of Korea

**Keywords:** COVID-19, chronic sinusitis, non-pharmaceutical intervention, interrupted time series analysis

## Abstract

Many countries have implemented non-pharmaceutical interventions (NPIs) to prevent the spread of COVID-19. However, the impacts of NPIs on the epidemiology and treatment of chronic rhinosinusitis (CRS) remain unclear. We analyzed 671,216 patients to investigate changes in the incidence rate and treatment frequency of CRS using Korean nationwide health insurance data between 2017 and 2021. The incidence rate (*p* < 0.001) and the number of outpatients (*p* < 0.001), patients hospitalized (*p* < 0.001), and patients prescribed antibiotics (*p* < 0.001) or steroids (*p* = 0.024) were significantly lower in the pandemic period than in the pre-pandemic period; however, the number of patients who underwent surgery was not different (*p* = 0.205). Additionally, the frequency of surgeries per patient was significantly lower in patients during the pandemic period (*p* < 0.001). In the interrupted time series analysis, the trends in the number of outpatients (*p* < 0.001), patients hospitalized (*p* < 0.001), patients who underwent surgery (*p* < 0.001), and patients prescribed antibiotics (*p* < 0.001) or steroids (*p* < 0.001) significantly changed after the onset of the COVID-19 pandemic. In summary, NPI implementation during the COVID-19 pandemic was associated with a reduction in the incidence and treatment of CRS.

## 1. Introduction

Coronavirus disease 2019 (COVID-19), caused by the severe acute respiratory syndrome coronavirus 2 (SARS-CoV-2), has had substantial impacts on public health. During the COVID-19 pandemic, many countries have implemented non-pharmaceutical interventions (NPIs), including face mask wearing, hand washing, social distancing, and closing schools, to mitigate the spread of SARS-CoV-2 infection [1,2]. As South Korea operates a National Health Insurance Service storing medical information from almost all of its citizens [3], it is the ideal country for investigating the effects of the COVID-19 pandemic on public health.

Chronic rhinosinusitis (CRS) is an inflammatory disease of the nasal cavity and paranasal sinuses with cardinal symptoms, including nasal obstruction, nasal discharge, facial pressure, and/or reduction in smell, persisting for more than 12 weeks [4]. CRS is a prevalent chronic inflammatory disease affecting 5.5% to 28% of the general population [5]. Endoscopic signs (including nasal polyps, mucopurulent discharges, and mucosal edemas) and changes in computerized tomography (CT) scans are frequently observed in CRS [5]. Histopathologically, CRS frequently presents with tissue remodeling, such as stromal edema, thickening of the basement membrane, squamous metaplasia, and goblet cell hyperplasia [6]. In addition, colonization by pathogenic bacteria and the presence of bacterial biofilm are frequently observed in the sinonasal mucosa of patients with CRS [7]. Patients with CRS may require medical treatments, including antibiotics, steroids, and endoscopic sinus surgery (ESS), if such treatments fail [4]. CRS can be initiated or aggravated by viral infections of the nasal and sinus mucosa [8]. The epithelial damage in CRS may lead to an increased invasion of respiratory viruses, subsequently aggravating the inflammatory response [9]. However, there are relatively few real-world data supporting the roles of respiratory viral infections in the pathogenesis of CRS.

The NPIs implemented against COVID-19 may prevent the spread of not only SARS-CoV-2 but also other respiratory viruses [10]. Given that respiratory viral infections act as a major driver of CRS aggravation, these interventions may elicit substantial changes in the frequency of medical visits and treatment of CRS. However, the impacts of NPIs implemented during the COVID-19 pandemic on the incidence and treatment of CRS remain unclear. In the present study, we aimed to investigate changes in the incidence and treatment of CRS during the COVID-19 pandemic in South Korea. To the best of our knowledge, this is the first nationwide study to investigate the effects of the COVID-19 pandemic on CRS.

## 2. Materials and Methods

### 2.1. Study Design and Data Collection

This study was approved by the Institutional Review Board (IRB) of the Yonsei University College of Medicine (IRB no. 4-2022-0919). All methods were performed in accordance with the relevant guidelines and regulations. The IRB of the Yonsei University College of Medicine waived the need to obtain written informed consent. This study considered the entire Korean population and the corresponding data were obtained from the Korean National Health Insurance claims database. The study period was from January 2017 to December 2021. This study used data from the National Health Information Database (M20220707001) of the Health Insurance Review and Assessment Service (HIRA). The HIRA database includes information on nearly all healthcare utilization of the Korean population (more than 50 million individuals). In Korea, the usage of every hospital procedure (including medical procedures and drug prescriptions) should be submitted to the Korean National Health Insurance Service by healthcare providers to request financial compensation, which is subsequently stored in the HIRA database. Not only the use of medical services but also other types of information, such as clinical information, diagnosis, and comorbidities based on the International Classification of Disease-10 (ICD-10) codes, are also available in the HIRA database.

The CRS incidence was identified according to the ICD-10 codes J32 (chronic rhinosinusitis) or J33 (nasal polyp). As in previous studies [11,12], CRS was defined as a case in which participants were treated more than twice (based on codes J32 or J33) and underwent head and neck CT scans (claim codes: HA401–HA416, HA441–HA443, HA451–HA453, HA461–HA463, or HA471–HA473). The inclusion criteria encompassed all patients with CRS in the study period. To select patients with CRS diagnosed for the first time, we set a washout period of one year. Thus, the exclusion criteria consisted of patients who visited the hospital with codes J32 and J33 in 2017. The primary endpoint of this study was outpatient visits, hospitalizations, surgeries, or prescriptions of medication (antibiotics or steroids) with J32 or J33 ICD codes. The information on ESS for CRS treatments was extracted using procedure codes. The data regarding the prescriptions of antibiotics and steroids were obtained using claim codes. The ingredients of the antibiotics and steroids are presented in Appendix A. The presence of comorbidities was also examined in patients with CRS based on ICD-10 codes, including hypertension (HTN) (ICD-10 codes: I10–I13, I15), diabetes mellitus (DM) (E11–E14), asthma (J45 and J46), allergic rhinitis (AR) (J301–J304), allergic conjunctivitis (H101), and atopic dermatitis (L20). The comorbidities were defined as those present within 1 year prior to the first claim of J32 or J33.

As the first COVID-19 case was reported on 20 January 2020 in South Korea [13] and the government-mandated NPIs on February 2020, the “Pre-pandemic period” was defined in this study as the period until January 2020, and the “Pandemic period” was from February 2020 to December 2021. Patients were categorized into the pre-pandemic or pandemic periods based on the first occurrence of J32 or J33. The details of the NPIs implemented in South Korea were provided by the Korean Disease Control and Prevention Agency and are summarized in Appendix A.

### 2.2. Statistical Analysis

The baseline characteristics were presented as means and standard deviations for continuous variables and as numbers and percentages for categorical variables. The incidence rates per 100,000 population were computed for different age- and sex-specific groups (in 10-year intervals) from the Korean general population (Korean Statistical Information Service, https://kosis.kr accessed on 8 November 2022).

The trend regarding the incidence and treatment of CRS by month during the pre-pandemic and pandemic periods was evaluated using a segmented regression analysis of an interrupted time series (ITS) [14]. ITS analysis performed in the present study is a highly recommended technique for analyzing data from a population-based interventional study [15]. Our segmented regression model is as follows:$$Y_{t}=\beta_{0}+\beta_{1}\times time_{t}+\beta_{2}\times transition\_pandemic_{t}+\beta_{3}\times transition\_pandemic_{t}+v_{t}$$
where $v_{t}$ is an autoregressive error model. A minimum of four parameters (β_0_, β_1_, β_2_, β_3_) are required for the segmented regression analysis. β_0_ is the baseline level of the incidence rate or healthcare utilization for CRS at time t = 0, i.e., January 2018; β_1_ is the average monthly change in the incidence rate or treatment of CRS before the transition to the pandemic; β_2_ is the immediate effect of the transition, i.e., the change in the level (drop or jump) of the incidence rate or treatment of CRS immediately after the transition, and β_3_ is the change in the slope (increase or decrease) of the incidence rate or treatment of CRS after the transition to the pandemic relative to the slope for the segment before the transition. Finally, β_1_ + β_3_ is the post-intervention slope of the segment after the transition, i.e., the average monthly change in the incidence rate of treatment after the transition to pandemic coding. The statistical test for this hypothesis verified the significance of the parameters of this regression coefficient.

A subgroup analysis was performed by categorizing the patients according to age as follows: 0–19; 20–39; 40–59; and ≥60 years. All tests were two-tailed, and *p*-values < 0.05 were considered statistically significant. The SAS Enterprise Guide (version 7.1; SAS Institute, Inc., Cary, NC, USA) was used for all statistical analyses. PROC AUTOREG syntax was used for the analysis.

## 3. Results

### 3.1. Baseline Characteristics of Study Subjects

In total, 1,150,227 cases of CRS were identified between 2017 and 2021. As we set a washout period of one year, patients visiting the hospital with codes J32 and J33 in 2017 (*n* = 479,011) were excluded; a total of 671,216 patients remained in the analysis. Correspondingly, 505,016 and 166,200 patients were identified for the pre-pandemic and pandemic periods, respectively. The proportion of male patients was significantly higher in the pandemic period (pre-pandemic vs. pandemic; 48.2% vs. 51.9%; *p* < 0.001). The mean age of the patients with CRS was also significantly higher in the pandemic period (pre-pandemic vs. pandemic; 47.68 ± 21.97 vs. 50.34 ± 21.49; *p* < 0.001). Accordingly, the frequency of patients aged >60 years was higher during the pandemic period. The utilization rates of tertiary and general hospitals were significantly higher during the pandemic period. When we examined the comorbidities in patients with CRS, the prevalence of HTN (pre-pandemic vs. pandemic; 30.6% vs. 34.9%; *p* < 0.001) and DM (pre-pandemic vs. pandemic; 19.8% vs. 23.2%; *p* < 0.001) was significantly higher in the pandemic period. In contrast, asthma (pre-pandemic vs. pandemic; 22.6% vs. 17.7%; *p* < 0.001), AR (pre-pandemic vs. pandemic; 76.8% vs. 67.2%; *p* < 0.001), and allergic conjunctivitis (pre-pandemic vs. pandemic; 15.3% vs. 15.1%; *p* = 0.011) were less prevalent in the pandemic period. The prevalence of atopic dermatitis was not significantly different before and during the pandemic (*p* = 0.858). The baseline characteristics of the study subjects according to the study period are presented in Table 1.

### 3.2. Changes in CRS Incidence and Number of Outpatient Visits and Hospitalizations during the COVID-19 Pandemic

The monthly mean incidence was 14.56 cases per 100,000 population during the pandemic period, which was significantly lower than the 39.49 cases per 100,000 population during the pre-pandemic period (*p* < 0.001; Table 2). However, the ITS analysis revealed no significant changes in the trend of decreasing CRS incidence after the onset of the COVID-19 pandemic (*p* = 0.110; Figure 1A; Table 3). The monthly number of total outpatients for CRS treatment significantly decreased after the onset of the COVID-19 pandemic (pre-pandemic vs. pandemic; 97,021.0 ± 21,975.2 vs. 77,667.9 ± 8873.0; *p* < 0.001; Table 2). In addition, the monthly number of total patients hospitalized due to CRS was significantly lower in the pandemic period than in the pre-pandemic period (pre-pandemic vs. pandemic; 2547.4 ± 436.1 vs. 2182.7 ± 187.7; *p* < 0.001; Table 2). Similar results were observed when we analyzed the annual frequencies of outpatient visits and hospitalizations per patient (Table 2). In the ITS analysis, the number of outpatients (estimated coefficient for level change β_3_ = −2709; 95% CI from −3318 to −2100; *p* < 0.001) and hospitalized patients (β_3_ = −64.53; 95% CI from −93.16 to −35.90; *p* < 0.001) due to CRS showed an increasing trend during the pre-pandemic period but began to decrease after the onset of the pandemic (Figure 1B,C; Table 3).

### 3.3. Changes in Treatment for CRS during the COVID-19 Pandemic

The monthly number of patients undergoing surgery for CRS was not significantly different between the pre-pandemic and pandemic periods (*p* = 0.205; Table 2). However, the ITS analysis showed that the trend of the monthly number of patients undergoing surgery significantly changed after the onset of the COVID-19 pandemic (β_3_ = −31.67; 95% CI from −47.68 to −15.66; *p* < 0.001; Figure 1D; Table 3). The annual frequency of surgeries per patient was significantly lower in the pandemic period than in the pre-pandemic period (pre-pandemic vs. pandemic; 0.12 ± 1.82 vs. 0.07 ± 0.69; *p* < 0.001; Table 2). In addition, we observed that the monthly number of total patients prescribed antibiotics (pre-pandemic vs. pandemic; 61,046.6 ± 14,264.3 vs. 40,881.9 ± 7334.3; *p* < 0.001) or steroids (pre-pandemic vs. pandemic; 21,261.4 ± 5007.1 vs. 18,694.4 ± 1948.2; *p* = 0.024) owing to CRS was significantly lower in the pandemic period than in the pre-pandemic period. The annual frequency of prescriptions of antibiotics (pre-pandemic vs. pandemic; 5.71 ± 21.33 vs. 1.65 ± 8.15; *p* < 0.001) or steroids (pre-pandemic vs. pandemic; 1.92 ± 11.62 vs. 0.71 ± 4.36; *p* < 0.001) per patient was also significantly lower in the pandemic period than in the pre-pandemic period (Table 2). The ITS analysis showed that the monthly number of patients prescribed antibiotics (β_3_ = −1630; 95% CI from −2053 to −1207; *p* < 0.001) or steroids (β_3_ = −564.49; 95% CI from −712.14 to −416.84; *p* < 0.001) showed an increasing trend in the pre-pandemic period but significantly decreased after the onset of the pandemic (Figure 1E,F; Table 3).

### 3.4. Subgroup Analysis According to Age

Next, we divided the patients into four groups according to age (0–19, 20–39, 40–59, and ≥60 years) and performed a subgroup analysis (Appendix A). We observed a significant reduction in the CRS incidence and number of outpatients and patients prescribed antibiotics during the pandemic in all age groups (Appendix A). Intriguingly, a significant difference in the monthly number of patients who underwent surgery between the pre-pandemic and pandemic periods was found in patients aged between 0–19 and 20–39 years but not in patients aged between 40–59 and ≥60 years (Appendix A). In addition, the monthly number of hospitalized patients and those prescribed steroids significantly decreased during the pandemic in patients aged 0–19, 20–39, and 40–59 years, but not in those aged 60 years or older (Appendix A).

## 4. Discussion

In the present nationwide study, we demonstrated that overall healthcare utilization for CRS significantly decreased during the COVID-19 pandemic. In particular, the number of outpatients and patients prescribed medications for CRS was substantially reduced during the COVID-19 pandemic, suggesting that NPIs implemented during the pandemic period might alleviate the initiation or aggravation of CRS by preventing respiratory viral infections. These findings may serve as evidence to support the effects of environmental factors on CRS.

Previous studies have described the critical roles of viral infections in the pathophysiology of CRS [8,16]. In particular, viral infections and secondary damage due to the host immune responses lead to disruption of the epithelial barrier [9]. As the epithelium is damaged, the increased susceptibility to secondary bacterial infection and persistent inflammation ultimately lead to CRS and/or trigger its exacerbation. In this regard, the reduction in respiratory viral and bacterial infections owing to NPIs may explain our results. As NPIs may prevent respiratory viral infections playing critical roles in the initiation or exacerbation of CRS, it seems plausible that NPIs have significant impacts on the incidence and treatment frequency of CRS. Additionally, the diminished exposure to air pollution by mask-wearing may influence the clinical course of CRS; in this context, accumulating evidence has suggested a positive association between air pollution levels and CRS aggravation [17].

Notably, we found that the patients with CRS in the pandemic period were older than those in the pre-pandemic period. It can be assumed that older patients with severe diseases requiring surgery may prefer to visit clinics more often than younger patients with mild diseases who choose to control their symptoms without medical consultation during the pandemic. This hypothesis may be supported by the fact that the monthly number of patients who underwent surgery did not change after the pandemic. The utilization rates of tertiary and general hospitals were also higher in the pandemic period than in the pre-pandemic period. In addition, the higher prevalence of HTN and DM in the pandemic period may be derived from the older ages of patients in the pandemic period relative to those in the pre-pandemic period.

Intriguingly, the prevalence of allergic airway diseases (including asthma and AR) partially sharing common pathophysiology with CRS [18,19] was significantly lower in patients with CRS in the pandemic period than in the pre-pandemic period. This result is in line with a previous study reporting decreases in the prevalence of asthma and AR in South Korean adolescents during the pandemic [20]. In addition, it was reported that mask-wearing reduces the symptoms of AR [21]. Because asthma and AR are closely linked to allergic sensitization [18,19], the wearing of face masks during the pandemic period may have blocked inhalation of respiratory allergens, leading to decreased medical visits for the treatment of allergic diseases. Similar to the case with CRS, the frequency of asthma exacerbation may also be reduced by NPIs that prevent respiratory viral infections, as such infections are a major driver of asthma exacerbation [22,23]. However, this interpretation should be cautiously considered, as we determined the presence of comorbidities based only on claim codes.

In the subgroup analysis according to age, we found that the monthly number of hospitalized patients and those who underwent surgery did not change in the elderly group during the COVID-19 pandemic. In addition, no significant changes in the number of hospitalized patients and those prescribed steroids were observed in patients aged 60 years. These findings may result from the high prevalence of fungus balls or nasal polyps, which frequently require surgical intervention, among older individuals [24,25]. However, the annual number of hospitalizations and surgeries per patient was significantly lower in the pandemic period than in the pre-pandemic period, indicating that NPIs during the pandemic may prevent post-operative recurrences requiring additional surgeries.

After the report of the first case of COVID-19 in January 2020, the South Korean government has implemented strict NPIs to mitigate the transmission of the virus. There have been restrictions on business hours, self-isolation measures, social distancing guidelines, mask mandates, nationwide lockdowns, school closures, and strong recommendations for vaccinations. These interventions act as physical barriers and alter environmental interactions, thereby blocking direct and/or indirect contact and invasions of respiratory pathogens. In addition, the gradual increase in hygiene awareness during the COVID-19 pandemic has also influenced the reduced rates of respiratory viral infections.

This study has several limitations. First, the diagnostic data relied solely on ICD-10 codes; detailed information on the diagnosis process (e.g., endoscopy and CT scan) and the clinical features of CRS (e.g., the presence of nasal polyps and disease extent/severity) are lacking. These clinical parameters are important factors in determining the need for surgery but were uncontrollable in this study. Second, social distancing and fear of the pandemic itself may have caused patients to hesitate to visit clinics and seek treatment despite having symptoms. Indeed, a previous study suggested that lockdown was associated with a reduction in the number of consultations in primary care [26]. In addition, similarities in symptoms between CRS and COVID-19 could potentially lead to even greater hesitations or limitations for patients in seeking medical attention. Therefore, the decline in hospital visits and overall healthcare utilization due to CRS might not accurately reflect a decrease in the prevalence of CRS. In this regard, the interpretation of the results should be exercised with caution. Further studies are needed to clarify the direct effects of NPIs on CRS. Further studies are needed to clarify the direct effects of NPIs on CRS. Third, the capacity of medical services for non-COVID-19 diseases may have been reduced during the pandemic. Fourth, information on socio-behavioral factors (e.g., alcohol use and smoking) and the roles of such factors were not examined in this study. Despite these limitations, the strengths of our study include a large number of patients, comprehensive analyses of real-world data, including the number of surgeries, antibiotics, and steroid prescriptions, and a longer post-pandemic study period of 2 years.

## 5. Conclusions

In summary, this nationwide investigation revealed that the incidence rate and healthcare utilization for CRS significantly decreased during the COVID-19 pandemic. Our results suggest that NPIs implemented during the COVID-19 pandemic may affect the epidemiology and treatment of CRS. These real-world data provide novel insights into the current understanding of CRS. Further long-term studies are needed to validate the direct effects of NPIs on CRS epidemiology and treatment.

## Figures and Tables

**Figure 1 jcm-12-06629-f001:**
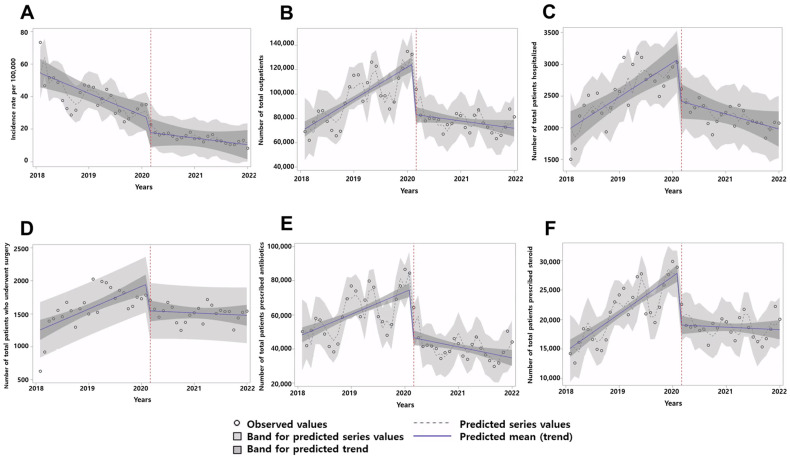
Interrupted times series analysis of chronic rhinosinusitis (CRS) incidence rates and healthcare utilization for CRS. Observed values and predicted trends in each primary endpoint parameter are depicted. (**A**) Monthly age- and sex-specific incidence rates per 100,000 population. (**B**) Monthly number of outpatients. (**C**) Monthly number of hospitalized patients. (**D**) Monthly number of patients who underwent surgery. (**E**) Monthly number of patients prescribed antibiotics. (**F**) Monthly number of patients prescribed steroids.

**Table 1 jcm-12-06629-t001:** Baseline characteristics of the study subjects.

Parameter	Total (*n* = 671,216)	Pre-Pandemic (*n* = 505,016)	Pandemic (*n* = 166,200)	*p*-Value
Sex				
Male	329,958 (49.2%)	243,637 (48.2%)	86,321 (51.9%)	<0.001
Female	341,258 (50.8%)	261,379 (51.8%)	79,879 (48.1%)	
Age, years	48.34 ± 21.88	47.68 ± 21.97	50.34 ± 21.49	<0.001
0–19	84,791 (12.6%)	67,652 (13.4%)	17,139 (10.3%)	<0.001
20–39	138,852 (20.7%)	106,021 (21.0%)	32,831 (19.7%)	
40–59	205,843 (30.7%)	155,861 (30.9%)	49,982 (30.1%)	
≥60	241,730 (36.0%)	175,482 (34.7%)	66,248 (39.9%)	
Hospital scale				
Tertiary	62,983 (9.4%)	43,927 (8.7%)	19,056 (11.5%)	<0.001
General	152,489 (22.7%)	112,547 (22.3%)	39,942 (24.0%)	
Clinic	454,936 (67.8%)	347,993 (68.9%)	106,943 (64.3%)	
Others	808 (0.1%)	549 (0.1%)	259 (0.2%)	
Comorbidities				
Hypertension	212,440 (31.7%)	154,419 (30.6%)	58,021 (34.9%)	<0.001
Diabetes mellitus	138,545 (20.6%)	99,983 (19.8%)	38,562 (23.2%)	<0.001
Asthma	143,574 (21.4%)	114,228 (22.6%)	29,346 (17.7%)	<0.001
Allergic rhinitis	499,558 (74.4%)	387,824 (76.8%)	111,734 (67.2%)	<0.001
Allergic conjunctivitis	102,330 (15.3%)	77,317 (15.3%)	25,013 (15.1%)	0.011
Atopic dermatitis	48,790 (7.3%)	36,726 (7.3%)	12,064 (7.3%)	0.858

Data are presented as number (%) or mean ± standard deviation.

**Table 2 jcm-12-06629-t002:** Incidence of CRS and healthcare utilization for CRS in the pre-pandemic and pandemic periods.

Parameter	Pre-Pandemic	Pandemic	*p*-Value
Monthly age- and sex-specific incidence rates *	39.49 ± 10.73	14.56 ± 3.18	<0.001
Monthly number of total outpatients	97,021.0 ± 21,975.2	77,667.9 ± 8873.0	<0.001
Annual frequency of outpatient visits per patient	8.73 ± 26.47	2.99 ± 11.74	<0.001
Monthly number of total patients hospitalized	2547.4 ± 436.1	2182.7 ± 187.7	<0.001
Annual frequency of hospitalizations per patient	0.23 ± 3.45	0.10 ± 1.40	<0.001
Monthly number of total patients who underwent surgery	1596.5 ± 315.6	1508.0 ± 127.0	0.205
Annual frequency of surgeries per patient	0.12 ± 1.82	0.07 ± 0.69	<0.001
Monthly number of total patients prescribed antibiotics	61,046.6 ± 14,264.3	40,881.9 ± 7334.3	<0.001
Annual frequency of antibiotics prescriptions per patient	5.71 ± 21.33	1.65 ± 8.15	<0.001
Monthly number of total patients prescribed steroids	21,261.4 ± 5007.1	18,694.4 ± 1948.2	0.024
Annual frequency of steroid prescriptions per patient	1.92 ± 11.62	0.71 ± 4.36	<0.001

* per 100,000 population. Data are presented as number (%) or mean ± standard deviation.

**Table 3 jcm-12-06629-t003:** Interrupted times series analysis of CRS incidence and healthcare utilization for CRS.

Parameter	Estimate	*p*-Value	Parameter	Estimate	*p*-Value
Monthly age- and sex-specific incidence rates *			Monthly number of total patients who underwent surgery		
Intercept β_0_	55.46	<0.001	Intercept β_0_	1224	<0.001
Baseline trend β_1_	−1.12	<0.001	Baseline trend β_1_	28.63	<0.001
Level change after intervention β_2_	−9.22	0.079	Level change after intervention β_2_	−395.63	0.001
Trend change after intervention β_3_	0.79	0.110	Trend change after intervention β_3_	−31.67	<0.001
Monthly number of total outpatients			Monthly number of total patients prescribed antibiotics		
Intercept β_0_	68,933	<0.001	Intercept β_0_	47,041	<0.001
Baseline trend β_1_	2206	<0.001	Baseline trend β_1_	1095	<0.001
Level change after intervention β_2_	−40,758	<0.001	Level change after intervention β_2_	−27,456	<0.001
Trend change after intervention β_3_	−2709	<0.001	Trend change after intervention β_3_	−1630	<0.001
Monthly number of total patients hospitalized			Monthly number of total patients prescribed steroids		
Intercept β_0_	1945	<0.001	Intercept β_0_	14,565	<0.001
Baseline trend β_1_	44.73	<0.001	Baseline trend β_1_	528.43	<0.001
Level change after intervention β_2_	−630.88	0.001	Level change after intervention β_2_	−8707	<0.001
Trend change after intervention β_3_	−64.53	<0.001	Trend change after intervention β_3_	−564.49	<0.001

* per 100,000 population. $\beta_{0}$ is the baseline level of the incidence rate or healthcare utilization for CRS at time t = 0, i.e., January 2018. $\beta_{1}$ is the average monthly change in the incidence rate or treatment of CRS before the transition to the pandemic; $\beta_{2}$ is the immediate effect of the transition, i.e., the change in the level (drop or jump) of the incidence rate or treatment of CRS immediately after the transition; $\beta_{3}$ is the change in the slope (increase or decrease) of the incidence rate or treatment of CRS after the transition to the pandemic relative to the slope for the segment before the transition.

## Data Availability

All data supporting the findings of this study are available within the main manuscript and the Appendix A or provided by the corresponding author (Min-Seok Rha) upon reasonable request.

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
