# Peer review of "Impact of Non-Pharmaceutical Interventions on the Incidence and Treatment of Chronic Rhinosinusitis during the COVID-19 Pandemic: A Nationwide Retrospective Cohort Study"

_jcm, 2023, doi:10.3390/jcm12206629_

Round 1
Reviewer 1 Report
The colleagues' work is interesting, but it requires some important changes.
The current wording of the title is vague and inaccurate: certainly, chronic rhinosinusitis has not changed as a pathological entity. Please fill in the title according to the objective of your study.
Please respect the journal's requirements regarding the abstract: the abstract should be a total of about 200 words maximum. The abstract should be a single paragraph and should follow the style of structured abstracts, but without headings: 1) Background: Place the question addressed in a broad context and highlight the purpose of the study; 2) Methods: Describe briefly the main methods or treatments applied. Include any relevant preregistration numbers, and species and strains of any animals used; 3) Results: Summarize the article's main findings; and 4) Conclusion: Indicate the main conclusions or interpretations. The abstract should be an objective representation of the article: it must not contain results which are not presented and substantiated in the main text and should not exaggerate the main conclusions.
Also, from the abstract written by you, neither the location of the study nor the number of included patients is understood. In this form, the summary is not attractive to readers.
The introduction must be completed and adapted in relation to your study, now it is formulated vaguel. Please study and cite doi: 10.3390/jcm10184110, doi: 10.3390/jcm9123973.
Materials and Methods, Study design and data collection. Please start with the consent of the ethics committee and/or the consent of the institutions involved.
Also, please clearly mention the inclusion and exclusion criteria.
Line 86 - codes J32 or J33, please explain.
Line 29 however, the number of patients who underwent surgery was not different (P = 0.205). Line 232 In particular, the number 232 of outpatients and patients who underwent surgery for CRS is substantially reduced dur- 233 ing the COVID-19 pandemic. Please explain clearly and correctly!
Line 309 To the best of our knowledge, this is the first nationwide study to investigate the effects of the COVID- 19 pandemic on CRS - please move this paragraph to the end of the introduction.
Please enter the conclusions in a separate part of the article and fill in strictly according to the purpose of your study.
Moderate editing of English language required
Author Response
The colleagues' work is interesting, but it requires some important changes.
- The current wording of the title is vague and inaccurate: certainly, chronic rhinosinusitis has not changed as a pathological entity. Please fill in the title according to the objective of your study.
Author response)
As the reviewer suggested, we revised the title from “Changes in chronic rhinosinusitis during the COVID-19 pandemic: A nationwide retrospective cohort study” to “Impact of non-pharmaceutical interventions on the incidence and treatment of chronic rhinosinusitis during the COVID-19 pandemic: A nationwide retrospective cohort study” in order to enhance clarity of scientific significance and objective of our study.
- Please respect the journal's requirements regarding the abstract: the abstract should be a total of about 200 words maximum. The abstract should be a single paragraph and should follow the style of structured abstracts, but without headings: 1) Background: Place the question addressed in a broad context and highlight the purpose of the study; 2) Methods: Describe briefly the main methods or treatments applied. Include any relevant preregistration numbers, and species and strains of any animals used; 3) Results: Summarize the article's main findings; and 4) Conclusion: Indicate the main conclusions or interpretations. The abstract should be an objective representation of the article: it must not contain results which are not presented and substantiated in the main text and should not exaggerate the main conclusions.
Author response)
In response to the reviewer's suggestions, we have made revisions to the abstract in accordance with the journal's guidelines.
- Also, from the abstract written by you, neither the location of the study nor the number of included patients is understood. In this form, the summary is not attractive to readers.
Author response)
We have added both the location of the study and the number of included patients to the abstract as follows:
“We analyzed 671,216 patients to investigate changes in the incidence rate and treatment frequency of CRS using Korean nationwide health insurance data between 2017 and 2021.”
- The introduction must be completed and adapted in relation to your study, now it is formulated vague. Please study and cite doi: 10.3390/jcm10184110, doi: 10.3390/jcm9123973.
Author response)
We have added the contents related to the pathophysiology of CRS and cited the references in the Introduction as follows:
“Histopathologically, CRS frequently presents with tissue remodeling, such as stromal edema, thickening of the basement membrane, squamous metaplasia, and goblet cell hyperplasia (doi: 10.3390/jcm10184110). In addition, colonization by pathogenic bacteria and the presence of bacterial biofilm are frequently observed in the sinonasal mucosa from patients with CRS (doi: 10.3390/jcm9123973).”
- Materials and Methods, Study design and data collection. Please start with the consent of the ethics committee and/or the consent of the institutions involved.
Author response)
As the reviewer commented, we have moved the sentences regarding the consent of the ethics committee to the beginning of “Study design and data collection”.
- Also, please clearly mention the inclusion and exclusion criteria.
Author response)
We clarified the inclusion and exclusion criteria as follows:
“As in previous studies [9,10], CRS was defined as a case in which participants were treated more than twice (based on codes J32 or J33) and underwent head and neck CT scans (claim codes: HA401 – HA416, HA441 –HA443, HA451 – HA453, HA461 – HA463, or HA471 – HA473). The inclusion criteria encompassed all patients with CRS in the study period. To select patients with CRS diagnosed for the first time, we set a wash-out period of one year. Thus, exclusion criteria consisted of patients who visited the hospital with codes J32 and J33 in 2017.”
- Line 86 - codes J32 or J33, please explain.
Author response)
Codes J32 and J33 represent “chronic rhinosinusitis” and “nasal polyp”, respectively. We added this information in the revised manuscript.
- Line 29 however, the number of patients who underwent surgery was not different (P = 0.205). Line 232 In particular, the number of outpatients and patients who underwent surgery for CRS is substantially reduced during the COVID-19 pandemic. Please explain clearly and correctly!
Author response)
Thanks for the helpful comment. We apologize for the misinterpretation in line 232 (of the original version of the manuscript) and have corrected the sentence as follows:
“In particular, the number of outpatients and patients prescribed medications for CRS is substantially reduced during the COVID-19 pandemic.”
- Line 309 To the best of our knowledge, this is the first nationwide study to investigate the effects of the COVID- 19 pandemic on CRS - please move this paragraph to the end of the introduction.
Author response)
We have moved the paragraph to the end of the Introduction.
- Please enter the conclusions in a separate part of the article and fill in strictly according to the purpose of your study.
Author response)
Following the reviewer's recommendation, we added the Conclusion section as a separate part of the article and outlined the conclusion of this study.
Moderate editing of English language required
Author response)
The manuscript's English was professionally edited by the "Editage" English editing service.
Reviewer 2 Report
The article is both interesting and of high scientific quality, particularly in its approach to big data analysis.
However, I have a few minor suggestions for improvement:
1. The abstract includes a conclusion that does not align with the study's aim. I recommend revising it to clearly present the study's most important findings.
2. Some statements, such as the impacts of the COVID-19 pandemic on chronic rhinosinusitis (CRS), lack clarity. For improved understanding, it would be helpful to specify whether you are discussing the epidemiology of CRS or its epidemiological impact on treatment or other interventions. This would make it clearer how CRS might be affected by COVID-19.
The English is ok.
Author Response
The article is both interesting and of high scientific quality, particularly in its approach to big data analysis.
However, I have a few minor suggestions for improvement:
- The abstract includes a conclusion that does not align with the study's aim. I recommend revising it to clearly present the study's most important findings.
Author response)
As the reviewer suggested, we revised the conclusion of the abstract to align with the aim of the study as follows:
“In summary, NPI implementation during the COVID-19 pandemic was associated with a reduction in the incidence and treatment of CRS.”
- Some statements, such as the impacts of the COVID-19 pandemic on chronic rhinosinusitis (CRS), lack clarity. For improved understanding, it would be helpful to specify whether you are discussing the epidemiology of CRS or its epidemiological impact on treatment or other interventions. This would make it clearer how CRS might be affected by COVID-19.
Author response)
We revised the statements from “the impacts of the COVID-19 pandemic on CRS” to “the impacts of NPIs implemented during the COVID-19 pandemic on the incidence and treatment of CRS” to provide clarity on how COVID-19 might affect CRS.
Reviewer 3 Report
This is an interesting and comprehensive study dealing with an intriguing and current topic.
The discussion is well developed according to results of the data collected from a large population.
As already discussed by Authors, fear and anxiety of patients about risks related to access to medical services during pandemic period may have heavily conditioned the real incidence of CRS requiring medical treatment or surgery.

Author Response
This is an interesting and comprehensive study dealing with an intriguing and current topic.
The discussion is well developed according to results of the data collected from a large population.
As already discussed by Authors, fear and anxiety of patients about risks related to access to medical services during pandemic period may have heavily conditioned the real incidence of CRS requiring medical treatment or surgery.
Author response)
As we already discussed limitations in the manuscript, patients' fear and anxiety regarding potential risks associated with accessing medical services during the COVID-19 pandemic could be a confounding factor when interpreting our results. Thus, further studies are necessary to elucidate the direct effects of non-pharmaceutical interventions on chronic rhinosinusitis (CRS) epidemiology and treatment. However, given that not only hospital visits but also the number of surgery and prescription of medication significantly changed after the onset of the COVID-19 pandemic, it can be inferred that there was a decrease in the incidence of CRS requiring treatment. Thanks for reviewing our manuscript.
Reviewer 4 Report
The study described has a strong explanatory power due to its large number of cases, and the study design also seems to me to be sensibly designed. The authors discuss the recommended protective measures as possible factors influencing the reduced number of cases of CRS. This seems logical in terms of content, but a large number of other possible influencing factors come into question.
The study has some weaknesses and limitations, but these are mentioned and discussed by the authors themselves and, in my opinion, adequately weighted.
Author Response
The study described has a strong explanatory power due to its large number of cases, and the study design also seems to me to be sensibly designed. The authors discuss the recommended protective measures as possible factors influencing the reduced number of cases of CRS. This seems logical in terms of content, but a large number of other possible influencing factors come into question.
The study has some weaknesses and limitations, but these are mentioned and discussed by the authors themselves and, in my opinion, adequately weighted.
Author response)
We agree with the reviewer that other factors may also affect the epidemiology and treatment of CRS. Therefore, further long-term studies are needed to validate the direct effects of non-pharmaceutical interventions on CRS epidemiology and treatment. We discussed this point in the manuscript. Thanks for reviewing our manuscript.
Round 2
Reviewer 1 Report
The manuscript was corrected according to the requested recommendations.